# Plants as Symbols of Power in the Achaemenid Iconography of Ancient Persian Monuments

**DOI:** 10.3390/plants12233991

**Published:** 2023-11-27

**Authors:** Giulia Caneva, Alessandro Lazzara, Zohreh Hosseini

**Affiliations:** 1Department of Science, University Roma Tre, 00146 Roma, Italy; giulia.caneva@uniroma3.it (G.C.); alessandro.lazzara@uniroma3.it (A.L.); 2National Biodiversity Future Center (NBFC), Università di Palermo, 90133 Palermo, Italy

**Keywords:** flower representation, phytoiconography, plant symbolism, Persian empire, Persepolis, rosettes, Susa, Iranian art, Zoroastrian religion

## Abstract

The art of the Achaemenid Empire flourished in Ancient Persia from the 6th to 4th centuries BCE, and featured stone-carved monumental structures adorned with recurring zoological and floral patterns. Such representations clearly had a symbolic meaning intimately connected to religious expression and the will of deities. Considering the lack of any comprehensive analysis of botanical features, we investigate the recurring plant patterns and the variety of plants depicted. An analysis of the documentation referring mainly to monuments in the two main capitals of Darius I, Persepolis and Susa, showed the presence of certain repetitive elements, such as the so-called rosettes (composed variously of Asteraceae capitula and *Nymphaea* flowers), palms (*Phoenix dactylifera*, the tree of life), pines, flowers or bunches and metamorphic elements. Some plants are described in this paper for the first time in the context of Persian iconography, such as *Mandragora officinalis* in offering scenes as a symbol of fertility and protection against evil spirits, *Pinus brutia* var. *eldarica* as a symbol of immortality and elevation to the gods, and the capitula of *Matricaria/Leucanthemum* as solar symbols. Further interesting elements include cf. *Myrtus communis* in some crowns and probably cf. *Ephedra* sp. in offering scenes. Achaemenid art was deeply influenced by the Zoroastrian religion of ancient Persia with its great attention to nature as well as by the nearby civilizations of the Mesopotamian area and Egypt. Most elements were also associated with psychotropic or medicinal attributes, which contributed to their position as symbols of power.

## 1. Introduction

Plant motifs represented in ancient paintings and archeological elements have mostly been understood in the context of their esthetic characteristics and their decorative function, even though ancient cultures saw all natural phenomena as being intimately connected with religious expression and the will of deities [1,2]. In this way, this symbolic system has various communicative functions, as an instrument for knowledge and construction of the objective world, and as an instrument of domination by establishing and legitimizing dominant cultures [3,4]. This intrinsic relationship between natural elements and artworks, linked to the symbolic value of the images that represent them, has already been shown for Egyptian [5,6,7], Greco-Roman [8,9,10,11,12], and other civilizations in the Mediterranean area and Ancient Near East [13,14,15,16,17,18]. In the same way, the role of zoological and floral motifs in Achaemenid art within the context of ancient Persian culture is considered key in communicating religious and cosmological beliefs, and they were often used to convey complex religious and philosophical ideas and express a wide range of symbolic meanings [19,20,21,22,23,24,25].

The Achaemenid Empire flourished in Ancient Persia between the 6th and 4th centuries BCE, beginning with the victory of Cyrus II the Great over Astyages the Mede in about 550 BCE until the conquests of Alexander the Great in 330 BCE. During this period, the Empire achieved rapid territorial expansion, stretching from Afghanistan to Anatolia and Egypt during the reign of Darius I the Great [19]. Achaemenid art incorporates elements from across this large territory and is known for its richness, elegance, and innovative techniques [26,27,28,29,30]. Achaemenid iconographic elements are characterized by the use of natural and ‘abstract’ stylized motifs, which also originated from typical geometric patterns. Their use can be traced back to the Zoroastrian religion, which emphasizes the importance of nature and the environment, as well as reflecting the Empire’s interest in horticulture and gardens, which were highly valued in Persian culture [31]. Many researchers have explored the meaning and symbolism of natural patterns found in Achaemenid iconography and the cultural exchanges that took place with neighboring civilizations [19,22,24,26,32].

However, we still lack a comprehensive understanding of the various functions and meanings within the iconography of ancient monuments, even those that represent the centers of power where every detail aims to convey a specific ideology. Although it has been shown that images played a powerful communicative role in ancient Roman culture [33] and in a wider iconological context [1,34,35,36,37,38,39], the role of plant-based motifs is often treated summarily and afforded little attention. The literature only traces general characteristics and focuses on dominant elements, such as lotuses and palms, while describing others simply as rosettes, flowers, or trees, while providing few taxonomic details [15,22,40,41,42], which are in some cases wrongly interpreted and often there is no exploration of why specific floral elements were chosen. In general, archeological studies often neglect the biodiversity behind such representations, whereas previous botanical investigations of several archeological monuments in the Mediterranean area have demonstrated its richness [10,12,43,44], for example, in the great attention that the ancients paid even to the minor details of plants and flowers [35,37].

Achaemenid art reflects the artistic traditions of the Iranian plateau, with its long and rich history dating back to the Bronze Age, but it was also influenced by the various civilizations that the Persian Empire encountered [45,46]. The most prominent of these seem to be from the Mesopotamian area, particularly the Neo-Assyrian Empire, which the Achaemenids conquered and absorbed into their empire, but also cultures from the ancient Near East, including the Elamites, Babylonians, and Egyptians [47,48]. The Achaemenids were, in fact, keen collectors of art and artifacts from nearby civilizations and often commissioned works from their artists and craftsmen. This influence can be seen in the use of hieroglyphic and cuneiform scripts, the depiction of sphinxes, and symbolic motifs, such as the winged disc [49]. However, we still lack a detailed understanding of the influence of such cultures on plant-based motifs. It would be useful to identify botanical elements more precisely in order to fully appreciate Iranian archeological sites, and hopefully some of these elements will be taken into consideration when planning the reconstruction of ancient landscapes and in museum activities aimed at interpreting the relationship between gods and nature [50].

This paper aims to employ a detailed analysis of motifs used in the visual language of Achaemenid monumental art, especially in royal palaces, placing them within their historical and cultural context in order to: (a) investigate common patterns and the variety of the plant elements used; (b) analyze the various meanings and functions of plant-based motifs: and (c) understand the influence on Achaemenid art of Ancient Persian religious traditions and those of other nearby civilizations.

Data are analyzed considering iconographical sources for plants and monuments, such as historical references for their symbolic attribution and previous archeological interpretations. Data of the nearby civilizations are also considered for understanding influences and origins of the floral patterns.

## 2. Material and Methods

### 2.1. The Dataset of Plant-Based Motifs

Given the abundance of surviving Achaemenid reliefs and artifacts, we focused our study on iconographic data derived from Achaemenid reliefs from Persepolis and Susa, founded by Darius I in 518 and 522 BCE, respectively, making them the primary sources for our research. We also looked at a limited selection of plant motifs in other Achaemenid sites, such as the relief of Bisotun (commissioned by Darius I in 520 BCE), and the Achaemenid royal tombs of Naghsh-e Rostam (522–330 BCE). Data for this research were collected from on-site surveys, online museum collections, archival resources, and bibliographic research. For the Palace of Darius I in Susa, we studied 16 reliefs, including 6 reliefs showing a winged lion, bull, or sphinx; 7 reliefs with a frieze of archers (31 archers, 17 of whom wore clothes decorated with plant motifs); and 2 reliefs with only floral motifs (1 geometrical pattern with rosette border and 1 palm pattern with palmette and rosette border). Images of the Persepolis reliefs were obtained through photographs taken by the authors at the archeological site and in the Tehran National Museum. Access to some parts of the site was limited, and in these cases, we used 294 images from the photographic archive of the Oriental Institute of Chicago, in the collection ‘Persepolis Terrace: Architecture, Reliefs, And Finds’ by Schmidt 1934–1939 (https://isac.uchicago.edu/collections/photographic-archives/persepolis/persepolis-terrace-architecture-reliefs-and-finds, accessed on 17 May 2023). Photographs of the Susa site was obtained from the brick panel and decorative brick collection of the Louvre Museum, Department of Oriental Antiquities (https://collections.louvre.fr, accessed on 17 May 2023). Rooms 307 and 308 house a display of archeological finds from Susa and a comprehensive collection of brick panels from the Palace of Darius I.

### 2.2. The Identification of Recurring Patterns and Related Plant Species

Plant motifs were later categorized according to their similarities and repetition in different parts of motifs, mainly by following previous surveys [14,15,22,24,32,40,51]. In keeping with previous classifications of plant patterns, we divided elements into the following categories: the so-called rosettes (stylized flowers with petals arranged in a star shape); single flowers, fruits, or bunches; palms and trees; parts of herbaceous plants; and metamorphic elements. Following the methodology for the identification of plants that we adopted in past studies in a similar artistic context [11,12], we considered several morphological data regarding the plant elements, such as structural features (size, plant architecture, and habitus, i.e., herbaceous, arboreous, or climbing), and since often only single parts of plants are shown, at least one of the following elements: the shapes and general structure of leaves (oval, truncate, elliptical, lanceolate, and linear; simple or compound, edges, and margins; and their arrangements on the stem) and the overall morphology of the flowers (number of petals, color, and symmetry), of inflorescences or their parts (e.g., ligulae of capitula), and of the fruits (typology, shape, size, and symmetry) or of other reproductive structures (e.g., the shape and size of cones, in the case of Gymnosperms). Considering the attention to detail paid by the ancients in their depictions of nature [35], we also decided to take certain smaller elements into consideration, such as the ligulae and bracts in the inflorescences or the sepals in fruiting structures. Data were also compared with botanical atlases and online databases of Mediterranean and Middle Eastern flora (Acta Plantarum (https://www.actaplantarum.org, accessed on 20 September 2023); Dryades Home (https://dryades.units.it, accessed on 20 September 2023); Kew Royal Botanical Garden (https://powo.science.kew.org, accessed on 20 September 2023); Global Biodiversity Information Facility (https://www.gbif.org, accessed on 20 September 2023); The World Flora Online (https://www.worldfloraonline.org, accessed on 20 September 2023); and Tela Botanica (https://www.tela-botanica.org, accessed on 20 September 2023). These morphological data were then compared with taxa compatible with the geographical area in question, as well as Egypt, Mesopotamia, and other surrounding areas of the Middle East. The dataset for such information was developed by Rechinger [52], Zohary [53], Djamali et al. [54], Dehshiri [55], and Hosseini et al. [56]. 

When uncertainties about morphological details made precise taxonomical identification difficult, we also considered the uses of morphologically compatible plants in ancient traditions, like medicine and rituals [50], as well as their most common symbolic attributes, but the analysis of such cases is conducted in the Discussion Section.

### 2.3. The Religious and Cultural Significance of the Depicted Plants in Relation to Nearby Civilizations

We also took account the Persian Empire’s cultural and religious background and the economic importance of plants, for which we referred to a number of sources relevant to the context of our study, mainly the Bundahishn, or ‘The Book of Creation’ [57], and other liturgical texts in the Avesta [58]. We also consulted bibliographical research related to the interpretation of religious texts [59,60,61,62,63,64]. 

We conducted a parallel analysis of plants’ symbolic value with regard to their depictions in ancient Near Eastern cultures ranging from the Eastern area of Egypt to Assyria, Babylonia, Lydia, and Ancient Persia. These data were useful in providing a more solid interpretation of the images as a whole. We conducted a comparative study by looking at plant motifs in reliefs and objects from 2000 to 200 BCE from Mesopotamia (Assyrian and Babylonian), Iran (Elamite and Medes), Egypt, Lydian, and other related cultures through the online museum collections of the Louvre (https://collections.louvre.fr, accessed on 17 May 2023), the Metropolitan Museum (https://www.metmuseum.org/art/the-collection, accessed on 18 May 2023), the British Museum (https://www.britishmuseum.org/collection, accessed on 18 May 2023), and the Oriental Institute of Chicago (https://isac.uchicago.edu/collections/collections, accessed on 17 May 2023), using the following keywords: ‘Assyria’, ‘Babylonia’, ‘Elamite’, ‘Medes’, ‘Lydia’, ‘Ancient Near East’, ‘Middle East’, ‘Mesopotamia’, and ‘Egypt’ (Figure 1). Regarding the symbolism of plant motifs in the art of these cultures, we consulted bibliographical studies and archeological reports (e.g., [13,16,18,47,65,66,67,68,69]).

## 3. Results

### 3.1. Recurring Patterns and Related Plants

Our analysis of Achaemenid iconography revealed a great amount of repetition of plant patterns and of single elements, and their biodiversity was limited to a few species. Most of them are relevant to the landscapes of the Middle East, North Africa, and the East of the Mediterranean Basin, whereas other characteristics of the specific landscapes of such geographical areas and plants were mainly selected considering their great pharmacological power. 

The recurring plant motifs from the various royal palaces in Persepolis and the Darius I Palace in Susa are shown in Table 1 and Figure 2. Further elements (mainly rosettes) are also derived from the relief of Bisotun and Achaemenid tombs of Naghsh-e Rostam. The data showed that the so-called ‘rosettes’ were mostly used in the borders of reliefs and were the most commonly repeated element. Rosettes are represented everywhere, including on doorframes, in the volutes and column capitals, on the collars of bulls, and on the clothing of archers and other figures.

From a morphological and taxonomical point of view, we identified two main types of rosettes, i.e., a radial arrangement of flowering elements similar to star and sun motifs, in the shape of the capitulate rosettes and lotus rosettes.

Capitulate rosettes can be recognized by their wide central yellow disk and white rays, which clearly resemble the typical inflorescence capitula of the Asteraceae family (Compositae) with yellow tubulate flowers in the center and numerous white ligulate flowers in the rays. They can be seen represented frontally (Figure 3a–c), in profile (Figure 3d), or from below (Figure 3e). Based on the morphological characteristics and the ring shape of the capitulate rosettes disk, we propose overall *Matricaria chamomilla* L., but other species of the family (e.g., *Leucanthemum vulgare*) can be supposed.

Lotus rosettes in the style of the Egyptian lotus (*Nymphaea lotus* L.) are recognizable by the smaller central elements and can appear in three morphological versions: showing a tetramerous radiate system, which corresponds to the early stage of flowering (Figure 4, stages 1 and 2); a radiate system characterized by a small central yellow disk with long white rays corresponding to mature flowering (Figure 4, stage 3); and a typical star arrangement, which corresponds to the transformation of the lotus ovary into a fruit (Figure 4, stage 4). In such stages are also the fruiting structures of *Nymphaea lotus* represented, such as the bracts, which surround the fruit (Figure 4, stage 5); the first stages of ovary maturation (Figure 4, stage 6), and the final formation of the fruit (Figure 4, stage 7).

The palm pattern (*Phoenix dactylifera* L.) is another recurring motif, used mostly at the beginning and at the end of a scene. The date palm appears often, where an Egyptian lotus and an emerging Asteraceae rosette are combined to form a palm (Figure 5). According to McDonald [15], the so-called ‘Mesopotamian palmettes’, which are formed by a pair of lateral volutes with a deltoid or rounded appendage fixed between them, do not belong to the palm plant, but represent a polypetalous flower (rosettes), shown in profile in Figure 5a. 

Among the arboreal elements, it is also possible to identify the presence of pines (*Pinus brutia* var. *eldarica* (Medw.) Silba.) in the Apadana staircase (Figure 5f), which were sometimes incorrectly identified as *Cupressus* [26,73], but have previously been recognized as pines by Ghirshman [74] and Roaf [75].

Ceremonial flowers, arranged in uniquely formed bouquets or, elsewhere, together with fruits, are common in ritual offering scenes, like those decorating the eastern staircases of the Apadana Palace in Persepolis, showing the hand-over-wrist gestures of nobles (Medes and Persians), or in the Audience Scene of the Treasury, where the most commonly depicted flower is the *Nymphaea*, which appears in various forms, sometimes together with mandrake fruits (*Mandragora officinarum* L.), and more rarely with *Cyperus papyrus* or *Ephedra* stems (Figure 6).

The pomegranate (*Punica granatum* L.) was identified in both forms of its initial blooming and in the form of fruit formation in the ancient reliefs of Xerxes in Persepolis. The representation of its fruit was used to adorn the parasol of the king (Figure 7d) and probably the apex of the royal scepter, which is at present in a bad state of conservation. The representation of its initial blooming seems evident in the plant bouquet, which is held in the hands of Xerxes in the Audience Scene (Figure 6f and Figure 7g), as also suggested by Llewellyn-Jones [48] and Darvishi [42]. Indeed, such representation arises from a combination of *Nymphaea’s* flower and *M. officinarum* fruits, which as the whole form the blooming pomegranate (Figure 7f). A combination of single elements forming a different representation would become famous in Arcimboldo’s style during the XVI century. 

The different species depicted are listed below along with a description of their taxonomical details, structure, chorology, ecology, recurrence, and diagnostic elements.
*Gymnospermae*

Cf. *Ephedra* sp. Family: Ephedraceae. *Common name*: Ephedra. *Biological form*: Chamaephytes. *Chorology*: The genus has a wide distribution in dry regions in both the Eastern and Western hemispheres. *Ecology*: Dry habitats. *Part of plants*: Stem. *Occurrence*: Bouquets in the Apadana staircase offering scenes (Persepolis). *Diagnostic elements*: The identification is uncertain due to the lack of precise diagnostic elements, but the linear shape of the bouquets makes this plant, which was prominent in Ancient Persian tradition, a likely candidate.

*Pinus brutia* var. *eldarica* (Medw.) Silba. *Common name*: Calabrian pine. *Family*: Pinaceae. *Biological form*: Phanerophytes. *Chorology*: The native range of this variety is Transcaucasia to NW Iran. *Ecology*: A drought-resistant tree that grows on dry rocky and stony slopes. *Part of plants*: Entire plants. *Occurrence*: Several trees on the Apadana staircase (Persepolis). *Diagnostic elements*: The genus *Pinus* is detectable by the typical cones, long acicular leaves, and scaly, fissured, patterned bark (its pyramidal shape led to it being mistakenly interpreted as a cypress). The species identification is based on a consideration of its morphological characteristics and the biogeographical distribution of the plant.
*Angiospermae*


*Dicotyledons*


*Nymphaea lotus* L. *Common name*: White Egyptian Lotus. *Family*: Nymphaeaceae. *Biological form*: Hydrophytes. *Chorology*: Africa. *Ecology*: Flooded lands in the wet tropical biome. *Part of plant*: Flowers in different stages of blooming and formation of fruits. *Occurrence*: Very common. *Diagnostic elements*: The shape of the flower, with many petals forming a radiate system around the central reproductive part (formed by numerous yellow stamens and carpels), sometimes represented in the tetramerous initial stage (all *Nymphaea* species have four sepals, 7–40 petals, 20–700 stamens, and 5–47 carpels). Some depictions show the fruits, culminating in a stigmatic disk resembling a large poppy capsule divided into equally spaced segments. Furthermore, on the top of the capitals, we can recognize a depiction of the carpel in its early stage of maturation and the formation of the stigmatic disk. We note that the Greeks first used the name ‘lotus’ in reference to a white waterlily (*N. lotus*), whereas Athenaeus [76] later used the same name for the blue waterlily (*N. nouchali* var. *coerulea*) [77]. Both *Nymphaea* species may be represented here, but we consider the first one to be the most relevant.

Cf. *Myrtus communis* L. *Common name*: Myrtle. *Family*: Myrtaceae. *Biological form*: Phanerophytes (shrub). *Chorology*: From Macaronesia to Pakistan. *Ecology*: Sunny places but with a certain humidity. Part of plant: Branches with leaves and fruits. *Occurrence*: Once, arranged in the form of a crown. *Diagnostic elements*: The shapes of the leaves, the long petioles of the fruits, and the elongated shapes of the berries (6–8 mm). The uses for crowns and other ceremonial uses appear widely in Mediterranean and European cultural traditions and mythology.

*Punica granatum* L. *Common name*: Pomegranate. *Family*: Lythraceae. *Biological form*: Phanerophyte. *Chorology*: NE Turkey to NW Pakistan. *Ecology*: Arid or semiarid environmental areas and wide adaptability to different ecological conditions. *Part of plant*: Floral bud and fruits. *Occurrence*: Rare (apical of king’s scepter). *Diagnostic elements*: Typical shapes of the fruit (balaustion, 5–12 cm), with a roundish ovary and acute segments as the remains of the petals, and typical blooming structures, sometimes very poorly preserved.

*Mandragora officinarum* L. *Common name*: Mandrake. *Family*: Solanaceae. *Biological form*: Hemicryptophyte (perennial herbaceous plant). *Chorology*: N Italy to NW Balkan Peninsula. *Ecology:* Open habitats, such as light woodland and disturbed sites. *Part of plants*: Fruits (3–5 cm). *Occurrence*: Common in ceremonial offering scenes. *Diagnostic elements*: Roundish fruits (berries) surrounded by long persistent sepals. The identification is reinforced by the ancient symbolic, medicinal, and magic values of the plant.

*Matricaria chamomilla* L. *Common name:* Chamomile. *Family:* Asteraceae. *Biological form:* Therophyte. *Chorology:* Temp. Eurasia to Indochina. *Ecology*: Grasslands; grows on all soil types and is resistant to cold. *Part of plant:* Inflorescence like a single flower (capitula). *Occurrence:* Widely distributed. *Diagnostic elements*: Even if capitula (1–2.5 cm) with numerous white ligulate flowers forming whitish rays and many yellow tubulate flowers in the center are characteristic of many genera in the family, such as *Bellis*, *Anthemis*, and *Leucanthemum*, we suggest this species to be highly probable in the case of the representation of a ring element that results from the centripetal growth of single flowers, or of the acute shape of the central disks, which results from the progressive growths of the tubulate flowers. Other species, especially *Leucanthemum vulgare*, are discussed later in this paper (Figure 3d).


*Monocotyledons*


*Phoenix dactylifera* L. *Common name*: Date palm. *Family*: Arecaceae. *Biological form*: Phanerophyte (up to 30 m). *Chorology*: Middle East, Mediterranean Area, and North Africa. *Ecology*: Thermophilus and heliophilous species, which need humidity in the soil. *Part of plants*: Entire plants (as skylines). *Occurrence*: Very common. *Diagnostic elements*: The features of this plant arise from the typical assemblage of lotus flowers forming the stem and by the half capitulate rosette forming the crown.

### 3.2. Religious and Cultural Significance of the Depicted Plants in Relation to Nearby Civilizations 

Plants and gardens were of particular importance in Achaemenid culture, and royal gardens were ‘an empire in miniature’ filled with plants, birds, and other animals from every area of the king’s dominion as an emblem of royal power [78,79]. It has been observed that ‘the idea of the king creating a fertile garden—displaying both symmetry and order—constituted a powerful statement symbolizing monarchic authority, fertility, legitimacy, and divine favor’ [48]. Achaemenid plant motifs, notably the capitulate rosette and lotus blooming, were inspired by Ancient Egypt and Mesopotamia (Figure 8). For instance, the capitulate rosette that adorned Babylonian reliefs during Nebuchadnezzar’s reign (605–562 BCE) also featured in Susa’s reliefs at Darius I Palace (Figure 8b,c).

However, the predominant Zoroastrian religion profoundly influenced Achaemenid art and its connection of human beliefs and nature [80,81]. Zoroastrianism makes reference to the flowers dedicated to various divinities and used in a variety of rituals, for example, in the Bundahishn, or ‘The Book of Creation’, every flower belongs to one of the Amahraspand, ‘beneficent immortals’, who form part of the retinue of Ahura Mazdā [57].

Table 2 show a synthesis of the symbolic and cultural significance of the plants depicted, such as the different ways in which they were interpreted by each of the civilizations connected with the Achaemenids.

## 4. Discussion

Achaemenids were known for their religious tolerance and syncretism, and this is reflected in their art, which was influenced by diverse cultural and religious traditions [24,102,119,120]. Although Achaemenid art was inspired by many older neighboring civilizations, it developed its own distinctive style that showcased a unique identity and the ability to transform foreign influences into something exclusively Achaemenian [30]. They carefully selected symbols able to establish a distinct representation of kingship aligned with their own vision of royalty [119,121]. In fact, a number of symbols that appear in Achaemenid art can be traced back to Zoroastrianism, with the most prominent during the early Achaemenid period being a human figure within a winged disc representing Ahura-Mazdā [63,94]. In Achaemenid reliefs (e.g., the Royal Tombs of Naghsh-e Rostam or Bisotun), coins, and the cylinder seal of Darius the Great, the king stands before a fire altar raising his hand towards Ahura-Mazdā in the winged disk. Ahura-Mazdā holds a ring symbolizing royal command [122] and a bracelet with rosette motifs that can be seen as a symbolic association with the sun (Figure 8). The omnipresent winged disk, borrowed from Egypt and Assyria [123], according to the Avesta, ‘contains the sun’ (Avesta, Yasna 32.2 [58]) and is clearly a symbol of the sun and Asha (order and right working) and simply represents the *Khwarnah* or ‘kingly glory’, specifically the solar radiance of Ahura Mazda that embodies the concept of good fortune [63,124]. 

Despite the representation of a relatively small number of species, we cannot avoid stressing the profound knowledge of natural phenomena in Achaemenid culture, as demonstrated in the attention paid to the selection of specific details and the depiction of specific phenological phases. In this context, we can explain the limited number of species because of their role as symbols of power. In fact, in order to achieve this, it was particularly important to transmit a clear and simple message that could be easily understood by the entire population without misunderstanding, while projecting the idea of supreme order. With this in mind, we can easily understand why we see the same elements constantly repeated in royal monuments, like the carved representations, the shape of columns and capitals, and many other architectural elements. Similarly, only a small number of animal species are depicted, including bulls, lions, griffins, and the metamorphic Sphinx symbolizing chaos, all of which had well-established symbolic meanings [21,125].

The great attention that Zoroastrians paid to nature [57] is indeed evident in the depiction of details and their associated communicative functions, where specific stages of flowering or ripening would have different meanings. Such representational choices prove that the authors were not only familiar with the morphology of the species but also with their physiological mechanisms of pollination and the formation of fruits and seeds, as demonstrated by the complex depiction of fertilization processes [57,60,126]. The depiction of fruiting carpels on the capitals of columns supporting symbolic animals does not seem to us to be coincidental, since they represent the final stage of blooming (Figure 3g and Figure 4, stage 6), whereas only the bracts of capitula, which represent the initial stages, are depicted at the bottom of columns (Figure 4, stage 1).

### 4.1. The Capitulate and Lotus Rosette Motifs as Solar Symbols with a Prominent Role in the Iconography of Assyrian and Egyptian Origins

The origin of the rosette motif [19,127] can be traced back to the ancient Near East, where it featured in Assyrian, Babylonian, and Egyptian art [13,32,40,51,128,129,130]. Some scholars have suggested that the rosette is a conscious revival of the age-old imagery of the goddess Ishtar/Inanna, a symbol of the link between kingship and fertility, and generally an allusion to fertility and abundance [51,131]. Most scholars, however, see it as a symbol of the sun and of its life-giving properties, and consequently, it was also associated with fertility, growth, and renewal [26,74,130] as well as magic virtue [26]. The sun’s daily transit through the heavens provides divine protection against darkness and the forces of evil; it is the source of divine fertility and oversees the cycle of birth, death, and rebirth [15,47,132].

Similarly, in Ancient Persia, the rosette symbol was also closely connected with Meher or Mithra, the most popular divinity in the Indo-Aryan world [84]. He was associated with the sun and with justice and played a part in Mehr or Mehregan ceremonies [133]. 

The connection of capitulate rosettes with the sun arises from the morphological features of such inflorescences, which are characterized by a central disc that mirrors the solar disc, and the circular arrangement of white ligulate flowers, which resemble the radiating rays of the sun. Due to this connection, this pattern symbolizes vitality, energy, and life-giving power, as the sun is often considered the source of all life on Earth.

Indeed, capitula with numerous white ligulate flowers forming whitish rays and yellow tubulate flowers in the center are characteristic of many genera, such as *Bellis*, *Anthemis*, and *Matricaria*, and in the case of Assyrian reliefs, some rosettes have previously been identified as daisies or marguerites (*Bellis* and *Anthemis*) [32,51,128]. In the case of the tomb of Tutankhamun, *Anthemis psedocotula* was proposed [87]. The consultation of the Zoroastrian Book of Creation, the Bundahishn, suggests *Leucanthemum vulgare* as the most suitable species, which is the symbol of Ard (good blessing), i.e., a true statement and a material reality that embraces all of existence [60]. However, in some cases, the morphological characteristics observed seem more reminiscent of *Matricaria chamomilla*, which also has a long pharmacological tradition that has been preserved in Iran to this day [134,135]. 

The lotus rosette was the most common motif in the art of Ancient Egypt [68,77,127,128,130], a choice that has been explained by the emergence of its flower from the waters. It was a symbol of purity and latent power [130]. Furthermore, since the *Nymphaea*’s flower generally opens at dawn and closes at sunset, it was easy to see it as associated with the sun. A further reason for the importance of *Nymphaea* can be found in its medicinal power, similar to that of the eastern lotus (*Nelumbo nucifera*) of the Ancient Aryans, famous for its psychotropic effects, which justifies the divine symbolic status that this plant already held in India’s prehistoric past [136]. Furthermore, *Nymphaea* species produce opiate alkaloids [137], and there is evidence that these psychotropic constituents were used among various populations in Ancient Egypt, India, and America to induce visions and euphoric states of mind [5,20,107,138,139,140]. It has also been suggested that their extracts, together with those of mandrakes, were employed by Egyptian healers in shamanic ceremonies, and an analysis of the ritual and sacred iconography of dynastic Egypt (represented on stelae, magical papyri, and vessels) indicates a profound knowledge of plant lore and altered states of consciousness. It seems that, through the power of plants, shamanic priests guided the souls of the living and dead and provided for the transmutation of souls into other bodies [5,138].

### 4.2. The Date Palm as a Reinterpreted Metamorphic Tree of Life in Middle Eastern Cultures

The importance of the date palm in this geographical context can be explained by its value as an economic plant and by its resultant symbolic attributes. Subfossil desiccated fruits and/or seeds have been recovered from excavations at Susa (Iran) and in other areas of the Middle East [141]. *Phoenix dactylifera* is one of the few arboreal species that is able to survive harsh conditions, as long as there is abundant underground water. This palm requires a long and intense hot summer with little rain and very low humidity during the period following pollination [142], and its ability to thrive in the arid Mesopotamian environment made it a powerful symbol of life in the midst of a challenging landscape [97].

The plant has been the subject of a complex chain of interlocking associations, including Ištar, fertility, and kingship [22], and it was also identified as the ‘Sacred Tree’ or ‘Tree of Life’ of the older Assyrian civilization, perceived as having apotropaic powers [30,95,143]. It has been suggested that, during the Assyrian period, the date palm and its ritual fertilization were so important to society that it must have led to the plant receiving the symbolic status of a sacred tree [141]. In Egypt, the date palm was a symbol of fertility and divinity, with Osiris and Ishtar portrayed together with palm leaves [18,136]. Later, these concepts were transferred into Greek, Roman, and Christian cultures, where the date palm continues to hold great symbolic importance. In different cultures around the Mediterranean and in West Asia, palms are defined culturally as ‘keystone species’, i.e., species whose existence and symbolic value are essential to the stability of a cultural group over time [144,145].

The metamorphic shape of date palms represents another example of media-specific programs in Achaemenid art [22]. Iranian kings had a special date palm known as the ‘Royal’ palm. It was special because, as soon as a tree died, another one grew out of its roots. Because of this, the date palm was associated with fertility, prosperity, and purity, and therefore, it was held to be an emblem of immortality and royalty among ancient Iranians and often used in Zoroastrian religious ceremonies, representing resistance to evil forces [146]. Pomegranates and date palms were sometimes planted together in Persian gardens as symbols of paradise [42], and Fire Temples often have date palms growing on their premises [147].

### 4.3. The Assyrian Origins of the Pine Tree’s Association with Gods and Immortality

The presence of the pine in Achaemenid iconography finds support from pollen data from Lake Parishan in southwestern Iran, which provides strong evidence that the Achaemenids deliberately planted pine trees, demonstrating the cultural significance of this tree [148,149]. 

Pine trees held a prominent position in the traditions and culture of Ancient Assyria, providing us with an insight into their associations with deities, the concept of immortality, and cultural assimilation [49,65,83]. In Assyria, the symbolic importance of the pine tree is exemplified by the reign of King Ashur-Nasirpal II (883–859 BCE). Historical inscriptions from this period emphasize the role of pine and cedar trees in the construction of the king’s palace, showing that these trees were not only building materials but also powerful symbols of royal authority [65]. Pine trees, although often overshadowed by their cedar counterparts, found their own symbolism in ancient Mesopotamian literature, particularly in the *Epic of Gilgamesh*, dating back to 2100 BCE. While the epic primarily centers around cedar trees, pine trees share their symbolic importance due to their close association. The quest for cedar in the epic is intertwined with themes of strength and immortality, making pine trees a symbol of endurance and eternal life in Mesopotamian culture, and the Sumerians and Babylonians incorporated the pine tree into their religious practices [150]. Archeological findings reveal that pine cones were used as offerings to the gods, and represented fertility and the renewal of life [18,151]. The Achaemenids incorporated and personalized artistic motifs and adopted the shapes and forms of pines from Mesopotamian art, but infused these symbols with their own cultural interpretations. They merged them with the qualities and symbolism of cypress trees, which also had symbolic importance for Zoroastrianism, acquiring a cosmic function as the tree of the good spirit that helps the believer ‘to ascend to heaven’ [98,152]. The pyramidal form of the cypress also reminds us of the flames that rise from earth to heaven, thus perfectly symbolizing the Zoroastrian doctrine itself [153]. 

### 4.4. Other Sacred Ceremonial Plants and Their Significance in Ritual Offerings

A number of different ceremonial plants offered to the king were represented in ceremonial rituals (in the case of the Apadana staircase, this was probably the New Year ceremony [26,154,155] or traditional epic events [156]).

A prominent role is held by the mandrake (*Mandragora officinarum*), which was associated with the creation myth. It is for this reason that, when Gayōmart (according to Avesta, the first human to worship Ahura-Mazda) died, a mandrake grew from his semen after forty years. In this belief system, human beings, or the first man and woman in the world, ‘Mashya and Mashyana’, respectively, were created from the dual root of this plant [104,105]. In Ancient Egypt, the mandrake was one of the most important flowers in gardens, and it appears widely in their imagery as a symbol of potency and sexual vitality [13]. It is likely that the New Kingdom knew of the effects of, for example, mandrake, poppy, and cannabis as narcotics, sedatives, and pain relievers as well as their mind-altering properties [107,140], and therefore used in ritual magic and healing [140]. It is possible that the portrayal of this plant in Achaemenid art might not strictly adhere to its factual dimensions. In fact, despite potential deviations from accurate dimensions, these representations prioritize symbolic significance over precise measurements, aiming to convey authority and grandeur within Achaemenid art, as influenced by Egyptian elements. Finally, its golden round fruits were also seen as a symbol of the sun [5]. 

In the case of cf. *Ephedra* sp., while the specific morphological characteristics of this plant may not be immediately evident, we know that Iranians have known and used it from as early as the Zoroastrian period. The ritual practices involving Haoma might have evolved in an area with different flora from the final settlement, prompting substitutions with locally available substances. Scholars have proposed various candidates for Haoma, including Ephedra, which Iranians have historically used [157]. According to Falk [158], Parsi-Zoroastrians used *Ephedra procera* (perhaps *E. sinica* or *E. vulgaris* [108]), which was imported from the Hari River valley in Afghanistan. This plant contains the drug ephedrine and was a source of an Iranian anti-fatigue drink called Haoma, which was used in rituals in Persepolis and became a traditional drink of immortality and longevity [159]. 

Pomegranates (*Punica granatum*), with their dietary and medicinal properties, have been used as symbols of human fertility in the ancient Near East and Mediterranean regions since the fourth millennium BCE [143]. Pomegranates featured prominently in Assyrian rituals and royal gardens [47,65], and are depicted in rock relief representations (e.g., the Tombs of Tutankhamun [47,49,65,160] in Egypt and Nimrud in Assyria [47,49,65]). Several historical studies have found evidence of large-scale pomegranate plantations [149], and archaeobotanical pollen from Persepolis as well as other historical documents, such as those in the Persepolis Fortification Archive (PFa33), demonstrate that it was the subject of intensive arboricultural activities in the heartland of the Achaemenid Empire [161]. In Ancient Persia, the pomegranate was connected to religious beliefs and myths [162] and was a symbol of Anahita, the goddess of water and of fertility. The pomegranate was a highly respected plant in Zoroastrianism and was planted in the courtyards of Fire Temples, while its twigs and seeds were used in certain Zoroastrian rituals and customs [110,163]

*Cyperus papyrus* is native to northern Africa [100] and does not naturally occur in Ancient Persia. Their depictions probably refer to the symbolism of *C. papyrus* from Egypt, like in the Palace of Darius I in Susa, where the combination of the lotus stalk (a symbol of Upper Egypt) and the papyrus (representing Lower Egypt) signifies the union of the two lands [24]. 

Finally, myrtle (cf. *Myrtus communis*) is described as an attribute of Ahura Mazda [164,165] in Zoroastrianism and specifically in the Bundahishn [57]. Its fragrance was likened to the scent of rulers [60]. Herodotus noted that Persians used it in sacrifices and to cover the ground during celebrations, as a symbol of regrowth and vitality in Ancient Iran [118]. Furthermore, myrtle has been recognized as being well-known as a medicinal plant since the Sumerian period [116,166] and continues to be prominent in modern ethnobotanical references, e.g., [167]. The significance of Myrtle transcends time and culture, with a rich history in the ancient Near East and Mediterranean regions [117], where it is symbolizes a variety of concepts. It represented immortality and eternity in Ancient Iran [115], Greece [168,169], and the Middle East [170]. Myrtle also symbolized authority, beauty, victory, and youth in Ancient Greek and Roman cultures [8,171,172].

## 5. Conclusions

This study emphasized the significance of plants associated with the representation of power in Achaemenid monumental iconography, confirming the substantial recurrence of rosette motifs, composed of either *Nymphaea* flowers or of Asteraceae capitula, as well as palms (*Phoenix dactylifera*), in a metamorphic combination. New data were presented in this paper regarding the depicted plant species, which have not been identified previously. Such plants were very clearly selected for their notable symbolic value as solar symbols connected to the idea of life and regeneration. This association was made in connection with their morphological and ecological features, but in some cases, it was also related to their pronounced medicinal and psychotropic properties. The influence of Near East civilizations and of Egypt can be seen clearly. Such knowledge will have substantial potential usefulness in the reconstruction of ancient landscapes and in the recognition of the value of nature in museum communication.

## Figures and Tables

**Figure 1 plants-12-03991-f001:**
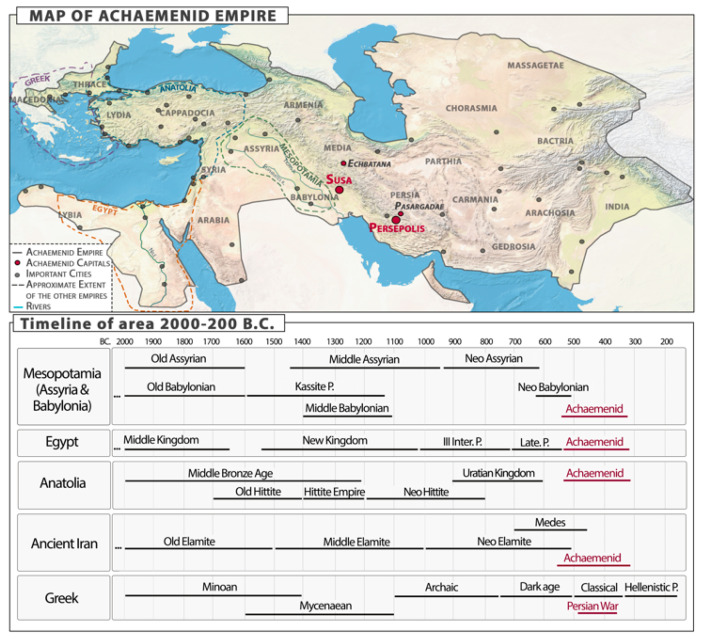
Map of the Achaemenid Empire and chronological and geographical contexts related to Achaemenid culture [70,71].

**Figure 2 plants-12-03991-f002:**
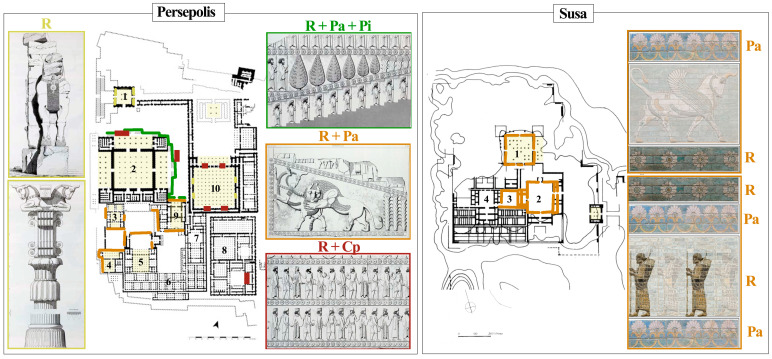
Distribution of the plant motifs in the royal palaces of Persepolis and Susa. Persepolis: (**1**) The gate of Nation, (**2**) Apadana Palace, (**3**) Palace of Darius, (**4**) Palace of Artaxerxes I, (**5**) Palace of Xerxes, (**6**) research center, (**7**) Harem of Xerxes, currently museum, (**8**) Treasury, (**9**) Council Hall, (**10**) Hundred Column Palace; Susa: (**1**) Apadana, (**2**) East Court, (**3**) Central Court, (**4**) West Court.

**Figure 3 plants-12-03991-f003:**
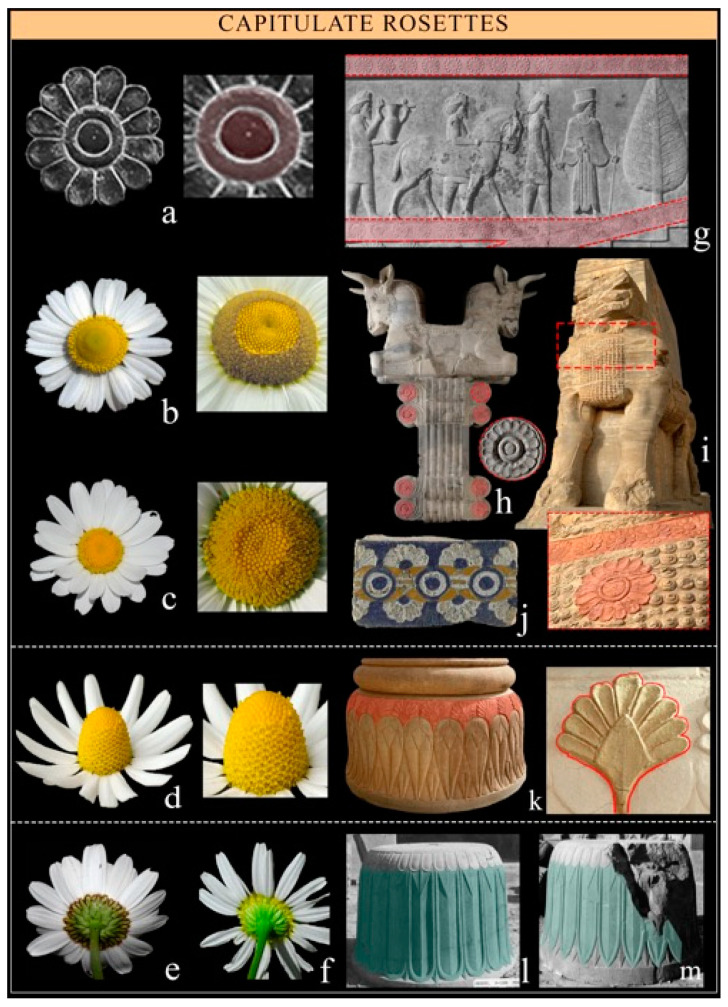
Capitulate rosettes (*Leucanthemum* and *Matricaria* sp.). (**a**) A 12-ray rosette pattern with the characteristic elements of the central disc (by Z. Hosseini); (**b**) *M. chamomilla* and its central disk (Tele-Botanica/CC-BY-SA 2.0 FR); (**c**) *Leucanthemum vulgare* and its central disk (the Royal Botanic Gardens, Kew); (**d**) *M. chamomilla* in profile (Tele-Botanica/CC-BY-SA 2.0 FR); the lower aspect of the capitula (**e**) *M. chamomilla* and (**f**) *L. vulgare* (World Flora Online); (**g**) rosettes in the scene borders of Apadana Palace in Persepolis (Courtesy of the Institute for the Study of Ancient Cultures of the University of Chicago); (**h**) rosettes in the capitula of the column in Susa (Musée du Louvre); (**i**) rosettes in the Sphinx’s neck, the Gate of All Lands at Persepolis (courtesy by G. Zangari); (**j**) glazed brick panel, Palace of Darius I, Susa (Musée du Louvre); (**k**) rosette in profile at the base column (by G. Zangari); and (**l**,**m**) rosette from the lower capitula in the base columns of Persepolis (Courtesy of the Institute for the Study of Ancient Cultures of the University of Chicago).

**Figure 4 plants-12-03991-f004:**
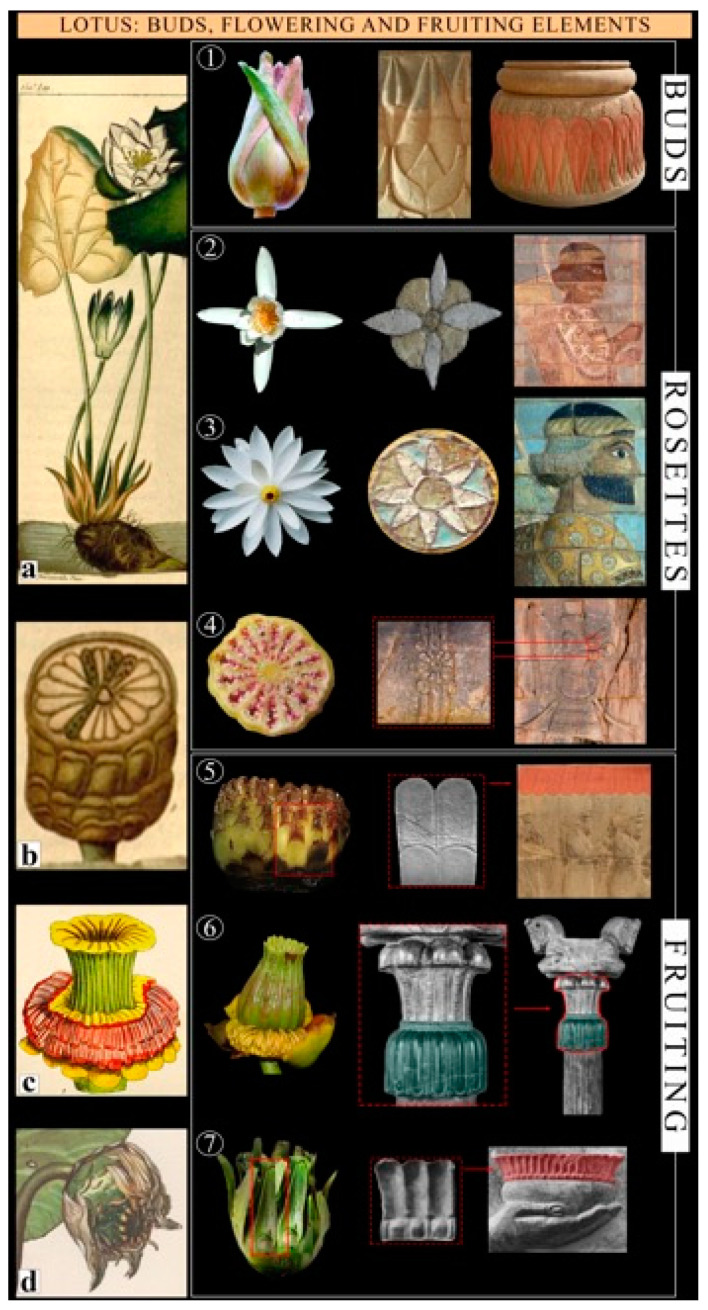
*Nymphaea* and Nymphaeaceae flower and fruiting elements. (**a**) The whole plant of *Nymphaea alba* (https://tsammalex.clld.org/parameters/nymphaealotus, accessed on 29 September 2023); (**b**) capsule with the star arrangement on the top and surrounding bracts (https://tsammalex.clld.org/parameters/nymphaealotus, accessed on 29 September 2023); (**c**) ovary maturation and formation of the upper disk (Botanical Magazine/Curtis); (**d**) bracts surrounding the fruit (Botanical Magazine/W. Curtis); (**1**) first stage of the plant as a bud [72] and its representation in some base columns of Persepolis (by G. Zangari); (**2**) initial flowering of the plant (African Plants/Stefan Dressler) and 4-ray rosette pattern in the dress of archers in Susa (the Metropolitan Museum of Art collection); (**3**) mature flowering by a small central yellow disk (Wikimedia/Midhun Subhash) and the lotus rosette in the dress of archers in Susa (Musée du Louvre); (**4**) the transformation of the lotus ovary into fruit shaped as a rosette [72] and its representation in the bracelets of Ahura-Mazda, the Xerxes Tomb of Naqsh-e Rostam (archive of the Pasargadae Research Center); (**5**) bracts of the first stage of fruit maturation (University of Wisconsin–Madison Botany Department Teaching Collection/Kowal, Robert R.) and its representation in the upper parts of the walls in the Apadana stairs of Persepolis (photo by Z. Hosseini); (**6**) intermediate stage of fruit maturation (http://www.plantsystematics.org/ accessed on 29 September 2023) and its representation in the columns of Susa (Musée du Louvre); and (**7**) final stage of fruit maturation with long bracts (http://www.plantsystematics.org/ accessed on 29 September 2023) and its representation in the Apadana stairs of Persepolis (by Z. Hosseini).

**Figure 5 plants-12-03991-f005:**
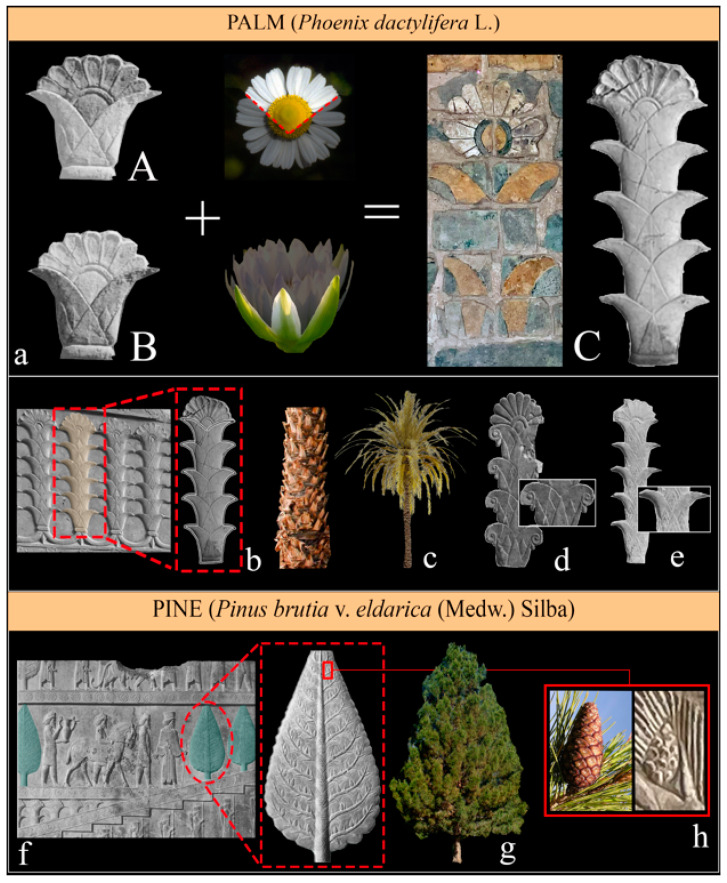
Palms and trees in Achaemenid art. (**a**) The combination of the Asteraceae rosette [A] with the Egyptian lotus [B] to create a date palm (*Phoenix dactylifera*) [C]; (**b**) detail of the date palm representation in Persepolis (Courtesy by G. Zangari); (**c**) the date palm and its stem on the left (the Royal Botanic Gardens, Kew); (**d**) a variation of the date palm, containing some with Egyptian details, from the Palace of Darius I, in the eastern stairway of Persepolis (Courtesy of the Institute for the Study of Ancient Cultures of the University of Chicago); (**e**) another variation of the date palm in the Palace of Darius I, in the southern stairway in Persepolis (Courtesy of the Institute for the Study of Ancient Cultures of the University of Chicago; (**f**) the Apadana stairs of Persepolis (Courtesy by G. Zangari); (**g**) *Pinus brutia* (Texas Tech University/Plant Resources); and (**h**) the pinecone as a characteristic element.

**Figure 6 plants-12-03991-f006:**
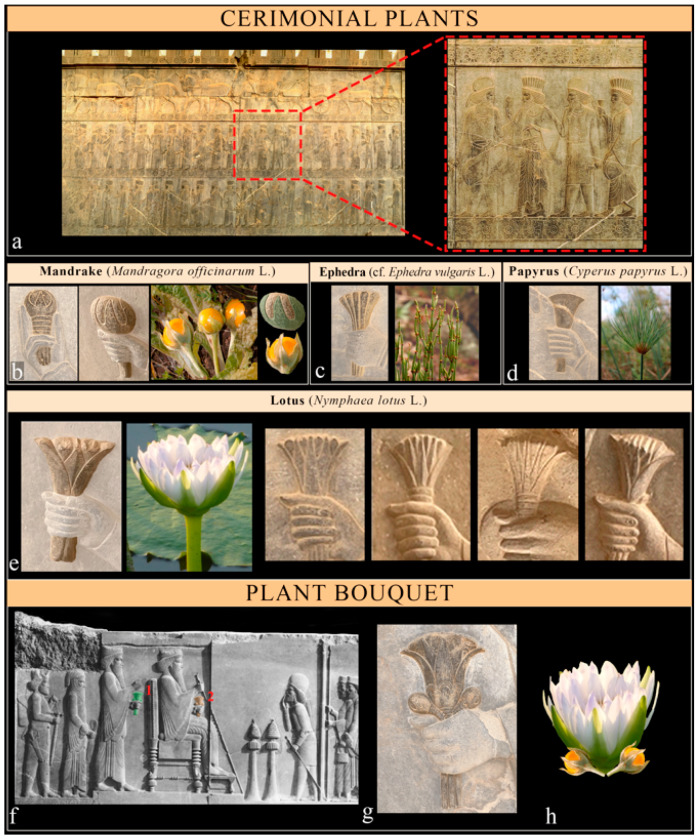
Ceremonial plants and plant bouquet. (**a**) Nobles holding ceremonial flowers at the Apadana stairs in Persepolis (all images courtesy by G. Zangari); (**b**) mandrake (*Mandragora officinarum*) and its representation in relief, fresh plants, and characteristic elements (Tagton’s Photography/DSCN4984); (**c**) ephedra (cf. *Ephedra vulgaris)* and its representation in reliefs and fresh stems (GBIF, Global Biodiversity Information Facility); (**d**) papyrus (*Cyperus papyrus*) and its representation in reliefs and its flowering structures (GBIF, Global Biodiversity Information Facility); (**e**) various representations of the lotus (*Nymphaea lotus*) and its blooming (African Plants/Stefan Dressler); (**f**) the Audience Scene in the Treasury, the eastern portico of the courtyard in Persepolis, depicting the king seated on the throne and his son/prince standing behind him, both holding (**g**) a bouquet of plants (Courtesy of the Institute for the Study of Ancient Cultures of the University of Chicago); and (**h**) the fresh combination of lotus and mandrake as represented in the relief related to bouquet 1—(representation of bouquet 2 in Figure 7g).

**Figure 7 plants-12-03991-f007:**
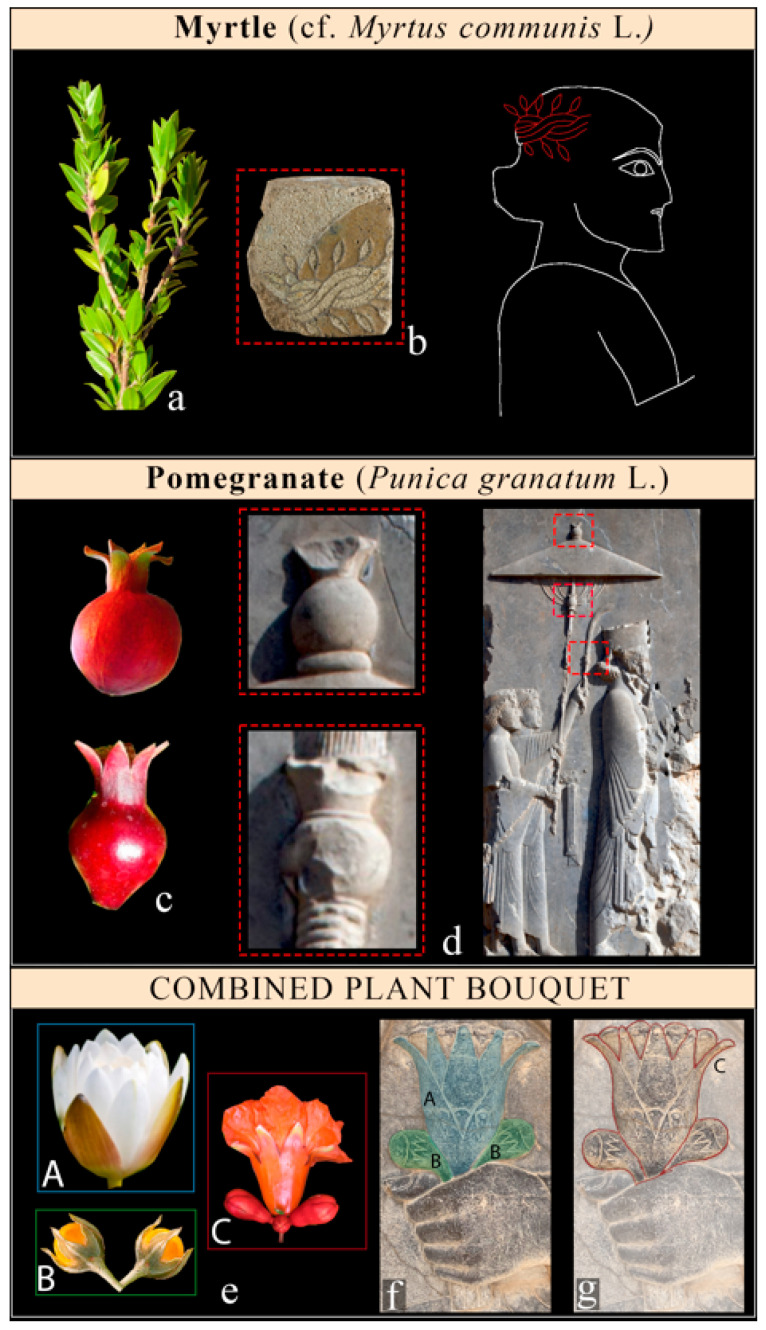
The other plants observed in the Achaemenid reliefs in Susa and Persepolis. (**a**) *Myrtus communis* (GBIF, Global Biodiversity Information Facility/Aldina Franco) and (**b**) its representation as two branches twisted together, probably in the head of archers (Musée du Louvre) and its position on the figure; (**c**) the fruit of *Punica granatum* (GBIF, Global Biodiversity Information Facility) and (**d**) its representation in the decoration of the king parasol at Xerxes’s relief in the doorway of his palace at Persepolis; and (**e**) the plant bouquet of the king (see bouquet 2 in Figure 5f) with two ways of looking it: (**f**) when separating the details as a combination of the lotus flower (A) with mandrake (B) and (**g**) when looking at the whole image as a general overview of pomegranate blooming (C).

**Figure 8 plants-12-03991-f008:**
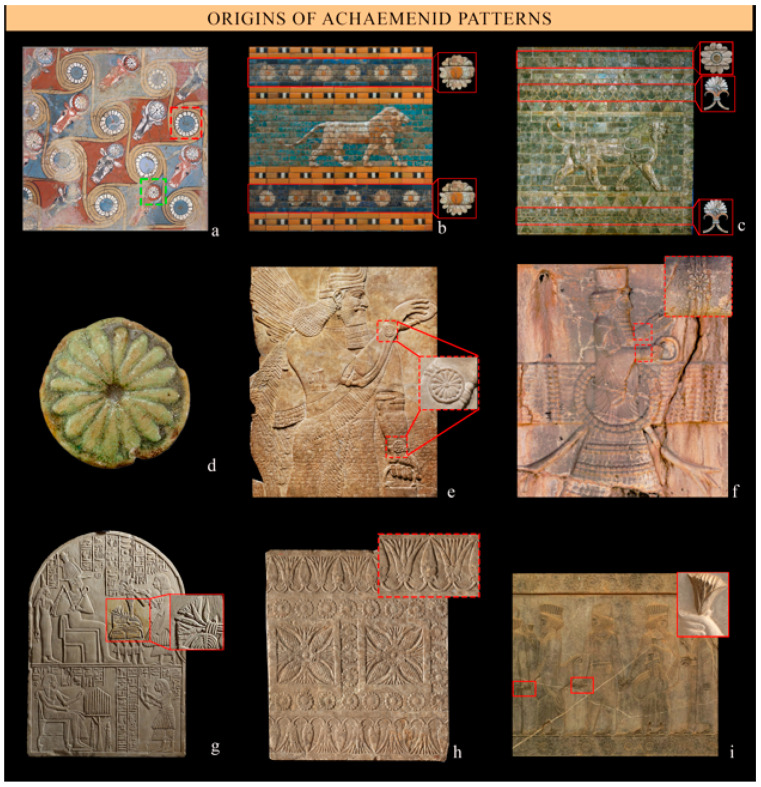
Influences of neighboring civilizations and origin of the Achaemenid patterns (on the left Egyptian representation; in the middle Mesopotamia; on the right Ancient Persia). Capitulate rosettes (in the red square): (**a**) ceiling decoration, Palace of Amenhotep III, Upper Egypt, 1390–1352 BCE (The Metropolitan Museum of Art collection); (**b**) glazed brick relief from reign of Nebuchadnezzar (605–562 BCE), Babylon, 6th century BCE (Berlin State Museums, Vorderasian Museum); and (**c**) brick façade of Susa, Palace of Darius I, 489 BCE (Musée du Louvre). Lotus rosettes: in the green square of (**a**); (**d**) ornament from the Palace of Ramesses II, Egypt, 129–1213 BCE (The Metropolitan Museum of Art collection); (**e**) the bracelet of the Ashur-nasirpal II reliefs, 865–860 BCE, Northwest Palace in Nimrud (The Metropolitan Museum of Art collection); and (**f**) the bracelet of the winged disk at the Xerxes Royal Tomb in Naghsh-e Rostam (Archive of the Pasargadae Research Center). Further lotus structures: (**g**) Egyptian stele, 1294–1279 BCE (Musée du Louvre); (**h**) threshold pavement slab with a carpet design, Neo-Assyrian, 700 BCE, Mesopotamia, probably from Nineveh (The Metropolitan Museum of Art collection); and (**i**) the frieze of nobles holding flowers in the Apadana stairways, Persepolis.

**Table 1 plants-12-03991-t001:** Recurrence of plant elements in the monumental remains of Persepolis (P) and Susa (S).

Elements		Total of Reliefs *		Rosettes (R)				Palm (Pa)		Pine (Pi)		Ceremonial Plants CP)	
				Capitulate R.		Lotus R.							
		P	S	P	S	P	S	P	S	P	S	P	S
Architectural Elements	Reliefs (R: Border lines; Pa: usually border; Pi and CP: variable)	170	45	124	6	-	6	37	9	17	-	13	-
	Wall tiles (ornaments)	4	12	4	10	-	-	-	-	-	-	-	-
	Columns (bases and capitals)	32	2	32	2	-	-	14	2	-	-	-	-
Human Figures	Dress (guards and archers)	-	31	-	-	-	17	2	-	-	-	-	-
	Crown (archer)	-	31	-	-	-	-	-	-	-	-	-	1
	Offering (in the hands of nobles, princes, and king)	+500	-	-	-	-	-	-	-	-	-	294	-
Sacred Animal Figures and Divine Representation	Bulls (collar)	29	6	26	2	-	-	-	-	-	-	-	-
	Lions (collar)	22	5	4	-	-	-	-	-	-	-	-	-
	Sphinx (collar)	8	3	4	2	-	-	-	-	-	-	-	-
	Winged disk (Ahura Mazda)	12	-	-	-	-	-	-	-	-	-	-	-

* The total number of evaluated reliefs in the sites.

**Table 2 plants-12-03991-t002:** Symbolic attributes of the main plants represented in Achaemenid monumental iconography and their associations with neighboring cultural areas.

Common Name	Ancient Cultural Area	Scientific Name	Represented Part	Symbolic Value
**Capitulate** **Rosettes**	Mesopotamia	*Hieracium pannosum* [65] and *Bellis perennis* [82]	Flower	Connected to the Goddess Inanna/Ishtar, representing kingship, fertility, and abundance [19,51,83]
	Ancient Iran	*Leucanthemum vulgare*	Flower	Emblem of the solar deity Mithra and a symbol of dynastic fertility [84]
	Egypt	*Picris coronopifolia* [85,86] and *Anthemis psedocotula* [87]	Flower	Emblem of the Sun God Ra [47], symbolizing the fertility of the Earth [88]
**Lotus Rosettes and other Lotus structures**	Mesopotamia	*Nymphaea speciosum* [65] and *N. alba* [40]	Flower	Symbolized rebirth and resurrection, regeneration, and eternal life [7]
	Ancient Iran	*Nymphaea* sp. [89]	Flower	Symbolized life and immortality [90] the creation, enlightenment and rebirth [41]
	Egypt	*Nymphaea lotus* [86,91], *N. caerulea* [5,85,86], and *N. alba* [20]	Flower	Myth of the birth of the God Horo [77], a symbol of resurrection and death–rebirth [92,93], the heraldic plant of Upper Egypt [85,94], and the sacred flower of the Nile God [47,86]
**Date palm**	Mesopotamia	*Phoenix dactylifera* [65]	Whole plant	Symbolized the sacred tree, offering fertility, prosperity [17], and apotropaic powers [95,96]
	Ancient Iran	*Phoenix dactylifera* [95]	Whole plant	Sacred to the God Mithra, symbolizing Mithraic power and authority [97]. Used in Zoroastrian religious ceremonies [98]
	Egypt	*Phoenix dactylifera* [15,99,100]	Whole plant	Connected with the Sun god [99], symbolizing fertility and resurrection [11,101]
**Pine**	Mesopotamia	*Pinus brutia* [65]	Whole tree	Connected to Ninurta, the god of agriculture, fertility, and warfare. Symbolized the god’s power and authority and eternal life [83]
	Ancient Iran	*Pinus brutia* var. *eldarica* [102]	Whole tree	Influence of Assyria
	Egypt	*Pinus pinea* cfr. [85]	Cone, tree	Connected to Osiris, the god of the afterlife and resurrection. Symbolizes rebirth and regeneration [85]
**Mandrake**	Mesopotamia	*Mandragora officinarum*	Fruit (berry)	Associated with fertility, magic and protection against evil spirits, love, and sexual potency [83]. Roots used as a potent analgesic/narcotic drug [103]
	Ancient Iran	*Mandragora turcomanica* [104]	Fruit (berry)	Associated with the creation myth [105]
	Egypt	*Mandragora officinarum* [87,106]	Fruit (berry)	Symbolized the sun in its golden round fruits and used for shamanic trance [5]; related to potency and sexual vitality, and roots and leaves were used medicinally [5,13,107]
**Papyrus**	Mesopotamia	----	----	----
	Ancient Iran	*Cyperus papyrus*	Stem	Symbolized Egypt
	Egypt	*Cyperus papyrus* [100]	Stem	Emblem of Ra [5] and heraldic plant of Lower Egypt [16,92]; home of the celestial divinity and mother Goddess Hathor [99]
**Ephedra**	Mesopotamia	*Ephedra sinica*	Stem	Stimulant and medicinal properties
	Ancient Iran	*Ephedra procera*, *E. sinica*, and *E vulgaris* [108]	Stem	Had a divine origin in Zoroastrianism and possessed healing and life-giving properties [109]
	Egypt	---	----	----
**Pomegranate**	Mesopotamia	*Punica granatum* [110]	Fruit	Cultivated to provide offerings for the cult of the New Year Festival [111]. Symbolized the deities of fertility, fecundity, and abundance and presented as offerings in religious rituals, with further medicinal uses
	Ancient Iran	*Punica granatum* [110]	Fruit	Emblem of Anahita (the Goddess of water and symbol of fertility) [41]. Associated with Mehr-Mitra and used in Zoroastrian religious rites [42]
	Egypt	*Punica granatum* [100,112]	Fruit	Symbolized love, prosperity, and fertility [112]. Used for dyeing textiles and leather [113,114], remedies [107], and the pomegranate wine production [91]
**Myrtle**	Mesopotamia	*Myrtus communis* [115]	---	Emblem of the Goddess Ishtar, hence becoming the plant of love [116,117]
	Ancient Iran	*Myrtus communis* [115]	---	Dedicated to Ahura Mazda [117], symbolizing immortality and eternity [118]
	Egypt	---	---	---

## Data Availability

Data are contained within the article.

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
