# Peer review of "Plants as Symbols of Power in the Achaemenid Iconography of Ancient Persian Monuments"

_plants, 2023, doi:10.3390/plants12233991_

Round 1

Reviewer 1 Report

Comments and Suggestions for Authors

This is an impressively well-organised approach to the identification of plants in ancient reliefs from Persia. The authors set out with great clarity the scope of the project, the way in which images have been grouped, and how identifications have been reached. The resulting data is then well-contextualised with wider understanding of the religious significance of the plants. The clarity of the evidence will make it easy for future researchers to evaluate these proposals (which, as ever with identifications of ancient plants, must always be tentative).

I do not have many suggestions for the text, which is well-written:

Throughout - consider supplying author epithets at first use (e.g. from POWO)

Bibliography -check for accuracy and spelling

Line 249, 312 etc - remove italics from var.

350 (and elsewhere) - would be useful to cite size of plant part shown. I had assumed mandrake fruits would be smaller but see they are in fact large enough to be the fruit shown in the image. Incidentally this seemed the least likely identification to me bearing in mind it is a purely European species.

571 - the source of Haoma remains unknown, it might be useful to add a note of caution to the identification (especially as the artistic identification is weak). See e.g. https://doi.org/10.1093/oxfordhb/9780190842642.001.0001

General

I would suggest referring to Germer's various publications, none cited in ms, for a more reliable view of which plants occurred in ancient Egypt than the cited papers by Embolden and others. Her 'Flora des pharaonischen Ägypten' is most useful for reassessing ancient claims.

It might also be good to refer to Sara Peterson's PhD thesis https://eprints.soas.ac.uk/26495/1/4479_Peterson.pdf which considers some of the same plants. Roses, Poppies and Narcissi: Plant Iconography at Tillya-tepe and Connected Cultures across the Ancient World

Reviewer 2 Report

Comments and Suggestions for Authors

The text is an approach to the use of plants depicted in the Persian Empire through the study of stone-carved representations in palaces and tombs of the Achaemenid period. Overall, it is a well-written text, sufficiently supported by documentation, which aims to expose the meaning of plants in the framework of the study. However, in my opinion, the manuscript contains some major problems of methodology and text structure that need to be amended or, alternatively, further explanation by the authors before it is suitable for publication. See the main comments here and these and other minor comments in the attached pdf.

Major comments

MATERIALS AND METHODS

In the methodology it is mentioned that when it is not possible to identify a species from the elements represented, written sources will be used. This is somewhat problematic for several reasons: 1) identification cannot be subject to a state of the art on a plant as this can change with new studies; 2), that we know of the use of a species does not mean that it is necessarily the plant represented; 3) representations are often idealisations or abstractions carved without having the real model as reference (the case of the lotus plant with the flower of an Asteraceae is an example of this). Therefore, I recommend that when there are no diagnostic elements that can clearly identify a species, the most precise botanical identification should always be used. In addition, when identifications are indicative, they should be preceded by "cf" e.g.: cf. Matricaria chamomilla L. Interpretations of plausible species base on written documents should be placed in the discussion section.

The above affect the following parts:

- Page 4, lines 148-151. Delete paragraph.

- Page 5, lines 199-202. The identification of Leucanthemum vulgare is an interpretation. Relocate in Discussion.

- Figure 3.7. The identification of this representation with lotus long bracts does not seem clear.

- Figure 6a. The identification of Myrtus communis from the plant elements is questionable.

The work does not contextualise the plants. It studies the botanical elements in isolation, and comments in a general way on the structures where they are located. There is no figure or information that allows us to know which panels are investigated, what is the theme of the panel, in which elements the plants appear (decoration of the scene, on a figure). Or, if applicable, if they are decontextualised representations. The absence of contextual information does not favour "a comprehensive understanding of the various functions and meanings within the iconography of ancient monuments" as this work mentions and tries to overcome.

The manuscript would benefit from the incorporation of one or more figures/tables/maps detailing the spatial distribution of the plant elements in the sites studied, in which buildings, rooms or panels, and which species correspond to which representations and characters: animate (humans, animals) or inanimate figures (objects, natural elements). I believe that a greater effort in this direction would enrich the manuscript, since it would provide a context to the elements studied, reinforcing the relationship between the meaning of each plant, and enhancing the discussion. This could be addressed either in the text or as Supplementary Material.

Results

The whole chapter 3.3 is not strictly results derived from the study but is a general overview consequence of the identifications and interpretations. Therefore, it makes more sense to relocate it as the first section of the discussion, as it is related to chapters 4.1, 4.2, and 4.3.

Discussion

This section needs a reorganisation to include section 3.3. 

Minor comments

There is no chapter 3.2. Either the title is missing or chapter 3.3. should be renumbered.

Merge Tables 2a and 2b, they contain the same information.

Round 2

Reviewer 2 Report

Comments and Suggestions for Authors

It is to be appreciated the effort made in the corrections made, to separate those interpretative phrases from the sections that should be reserved for the presentation of information in this new version.

Regarding the comment on my point 3 included in the first revision, I don't think we have different points of view necessarily. In general terms, my comment was that just as there are very faithful representations of plants, there are also imitations, emulations or copies of all or part of plant characteristic elements, which even go as far as abstracting the original model. In order to go deeper into this in future works, I would recommend reading publications on the concept of skeuomorphism, which is the direct and imitative relationship between two elements in different materials, so it is necessary to study both the characteristics of the imitated objects and the material and social context of the copy in order to understand the representations.

There are still some elements that need revision but can be remedied during the editing process.

Representation 7b, does not correspond to the figure of the archers on the right. Those Susa archers wear a clothe with the fillet of a rope on their heads. A cord headband very common among Near Eastern ancient cultures, not twisted plant branches. Please review this information throughout the text. A schematic representation such as 7b can be a myrtle or olive wreath (even more common than myrtle in this type of representations), or other species that can be easily twisted, e.g., Salix

There are still some typos that can be corrected, for example:

Figure 3: € should be (e).

Chapter 3.2. is still missing from the text. Please revise chapter numbering.
